# Ab Initio Elucidation of the Nature of the Bonding of Tetrahedral Nitrides (BN, AlN, GaN, and InN), Hexagonal BN, and Graphene

**DOI:** 10.3390/ma18122875

**Published:** 2025-06-18

**Authors:** Pawel Strak, Konrad Sakowski, Pawel Kempisty, Izabella Grzegory, Agata Kaminska, Stanislaw Krukowski

**Affiliations:** 1Institute of High Pressure Physics, Polish Academy of Sciences, Sokolowska 29/37, 01-142 Warsaw, Poland; konrad@unipress.waw.pl (K.S.); pkempisty@unipress.waw.pl (P.K.); izabella@unipress.waw.pl (I.G.); kaminska@ifpan.edu.pl (A.K.); 2Institute of Applied Mathematics and Mechanics, University of Warsaw, 02-097 Warsaw, Poland; 3Institute of Physics, Polish Academy of Sciences, Aleja Lotnikow 32/46, 02-668 Warsaw, Poland; 4Faculty of Mathematics and Natural Sciences, School of Exact Sciences, Cardinal Stefan Wyszynski University, Dewajtis 5, 01-815 Warsaw, Poland

**Keywords:** boron nitride, graphene, tetrahedral nitrides

## Abstract

Recent measurements of the band properties of AlN and GaN by fluorescence yield absorption and soft X-ray emission spectroscopies revealed that their valence band (VB) is composed of two separate subbands. The upper VB subband of GaN is composed of gallium *sp* and nitrogen *p* orbitals; the lower subband consists of metal *d* and nitrogen *s* orbitals. These findings were confirmed by extensive ab initio simulations. These results are not consistent with the standard tetrahedrally coordinated semiconductors, which are bonded by *sp*^3^-hybridized orbitals of metal and nonmetal atoms. The new analysis techniques and ab initio simulations create a new picture, allowing the calculation of overlap integrals to determine the bond order in these crystals. According to these results, bonding occurs between resonant *p*-states of nitrogen and *sp*^3^-hybridized metal orbitals in tetrahedral nitrides, allowing tetrahedral symmetry to be maintained. A similar resonant bonding mechanism is observed in hexagonal BN, where the *p* orbitals of nitrogen create three resonant states necessary for maintaining the planar symmetry of the lattice. In addition, nonresonant π-type bonds in BN are created by the overlap of $p_{z}$ orbitals of boron and nitrogen. BN bonding differs from that in graphene, where carbon states are fully *sp*^2^-hybridized. Additionally, π-type bonds in graphene have no ionic contributions, which leads to the formation of Dirac states with linear dispersion close to the *K* point, closing the band gap.

## 1. Introduction

Boron nitride (BN) belongs to several drastically different families of compounds, depending on its lattice properties: hexagonal structures belong to 2D materials, while tetrahedral structures belong to III–V semiconductor compounds [1]. These families differ in structural, mechanical, electrical, and optical properties. Hexagonal BN is a soft material, chemically resistant to many chemically aggressive agents, which is related to the strong chemical bond between boron and nitrogen atoms. This material is available in a number of different forms, including single crystals of large area and low thickness, flakes separated from these large crystals (also in single crystalline form), grains, or ceramics sintered from small grains. Tetrahedral BN exists in two different symmetries: cubic and wurtzite [2]. Both tetrahedral BN crystals are very hard—second only to diamond on the hardness scale.

Hexagonal BN has already been used in a number of lucrative applications [3]. Due to its thermal and chemical resistance and ease of manufacturing, BN ceramics are widely used in crystal growth technology. On the other hand, it is also widely used as a lubricant or even in the cosmetics industry [3]. Cubic BN, known as borazon, is used in mechanical tools, especially for cutting iron-based materials, where diamond is adversely affected, particularly at elevated temperatures due to friction [4]. Wurtzite BN is extremely hard to obtain and, thus, is not applied in technology.

In recent years, hexagonal BN has been intensively investigated with the prospect of future optoelectronic and electronic applications [5]. This provides an alternative to other nitrides, which are predominantly synthesized in tetrahedral structures—either hexagonal wurtzite or regular zinc blende. The stable form is wurtzite for the three compounds: AlN, GaN, and InN [6,7,8]. These nitrides are currently the basis of rapidly developing optoelectronics and electronics [9,10,11,12,13]. These applications were awarded the Nobel Prize in 2014 [14] and are expected to continue developing rapidly [15].

At the other end of the scale is graphene, a two-dimensional material that has been investigated over the past 15 years [16,17,18]. Despite high expectations and extensive research efforts, the promised applications have not yet been realized [19]. Nevertheless, graphene and other 2D materials remain a focal point of intensive research due to their promising future applications. One of the most promising is energy storage in a graphene–boron nitride (C-BN) capacitor structure [20]. At present, this is the most promising direction in terms of both the potential impact and probability of success. Therefore, the C-BN combination is of special interest, also from the point of view of applications.

An important fundamental issue is related to the bonding in these materials. The standard theory of semiconductors—both elemental and most III–V compounds—has provided a successful picture of bonding via ${sp}^{3}$-hybridized metal–nonmetal bonds, involving both covalent and ionic bonding [21,22]. Despite its tetrahedral structure, bonding in wurtzite GaN includes hybridized ${sp}^{3}$ orbitals of gallium and separate *s* and *p* orbitals of nitrogen. In addition, the bonding of *d* orbitals of gallium involves overlaps between gallium *d* and nitrogen *s* orbitals. The higher-energy bonding involves hybridized ${sp}^{3}$ orbitals of gallium and *p* orbitals of nitrogen. This creates two separate subbands in the valence band (VB), which were measured experimentally using soft X-ray spectroscopic measurements by Magnuson et al. [23]. Ab initio calculations confirmed the existence of these two subbands and helped identify the nature of the bonding [23,24,25]. In the case of AlN, subband separation was also identified, with the lower subband due solely to nitrogen *s* orbitals [23]. The subject of the present paper is the determination of bonding in wurtzite and hexagonal nitrides and its comparison to the bonding in graphene. This paper begins with a presentation of the calculation procedure, followed by the main results, which are summarized at the end.

## 2. Calculation Procedure

The density functional theory-based ab initio calculation procedure implemented in the SIESTA shareware package was used in the simulations of group III metal nitrides and graphene. The basic software was developed with the support of the Spanish Initiative for Electronic Simulations with Thousands of Atoms (SIESTA). The software solves a set of Kohn–Sham one-electron nonlinear equations [26]. The result is a set of eigenfunctions that are linear combinations of finite-radius atomic orbitals from a predefined functional set [27,28]. The angular dependence is expressed using harmonics. In most cases, these are *s*, *p*, and *d* polynomials of sine and cosine functions of spherical angles. For precision, the *s* and *p* orbitals of metal and nitrogen atoms are represented by triple-zeta functions. Additionally, the *d*-shell electrons are incorporated into the valence electron set for gallium and indium. The *d* orbitals are reduced to a single zeta function. The set of eigenfunctional is drastically reduced by applying Troullier–Martins pseudopotentials [29,30]. The integration in k-space is replaced by a direct summation over a Monkhorst–Pack grid of $\left(7\times 7\times 7\right)$ points [31]. Additionally, the GGA-PBE (PBEJsJrLO) functional is parameterized using β, µ, and κ values determined by the jellium surface (Js), jellium response (Jr), and Lieb–Oxford bound (LO) criteria [32,33]. In the nonlinear matrix solver, the self-consistent field (SCF) loop is terminated when the difference between any two consecutive iterations of the density matrix is less than 10^−4^. A final representation in the real space in grid form was designed for calculating the multicenter overlap integrals, which are controlled by an energy cutoff value set to 410 Ry. This can be translated into a grid spacing of 0.08 Å in real space.

The lattice parameters of the bulk wurtzite boron nitride, obtained from ab initio calculations, are $a_{w-BN}^{DFT}=2.5417\;Å$ and $c_{w-BN}^{DFT}=4.2019\;Å$, while X-rays give $a_{BN}^{exp}=2.550\;Å$ and $c_{BN}^{exp}=4.227\;Å$ [2]. DFT lattice data calculated for wurtzite AlN are $a_{AlN}^{DFT}=3.1126\;Å$ and $c_{AlN}^{DFT}=4.9815\;Å$, in agreement with X-ray data from bulk aluminum nitride wurtzite crystal: $a_{AlN}^{exp}=3.111\;Å$ and $c_{AlN}^{exp}=4.981\;Å$ [6]. The calculated values for wurtzite GaN are $a_{GaN}^{DFT}=3.1955\;Å$ and $c_{GaN}^{DFT}=5.2040\;Å$, in agreement with X-ray results: $a_{GaN}^{exp}=3.1890\;Å$ and $c_{GaN}^{exp}=5.1864\;Å$ [7]. For InN, these data are $a_{InN}^{DFT}=3.5705\;Å$ and $c_{InN}^{DFT}=5.7418\;Å$. They are in good accordance with the experimental data for wurtzite InN: $a_{InN}^{exp}=3.5705\;Å$ and $c_{InN}^{exp}=5.703\;Å$ [8]. The lattice constants for hexagonal boron nitrides are $a_{h-BN}^{exp}=2.504\;Å$ and $c_{h-BN}^{exp}=6.661\;Å$ [34]. The theoretically obtained values are $a_{h-BN}^{DFT}=2.51\;Å$ and $c_{h-BN}^{DFT}=6.82\;Å$. Finally, the experimental values for graphene AB stacking (Bernal) are $a_{C}^{exp}=2.461\;Å$ and $c_{C}^{exp}=6.709\;Å$ [35]. Note that the nitrides are typically affected by the high density of point defects, both native [36] and substitutional atoms [37], which influence lattice constants. For example AlN could be affected by the annealing process, either during or after growth, due to the escape of highly volatile nitrogen [37]. Additionally, aluminum (Al) is extremely easy to oxidize. Thus, aluminum nitride (AlN) usually contains substitutional oxygen, which creates a mass defect in the lattice [37].

The standard approach underestimates the bandgap when determining the energy of quantum states [32]. Therefore, the GGA-1/2 approximation of Ferreira et al. was used [38,39]. This approximation yields good agreement with the optically measured bandgaps [32,39,40,41,42,43,44,45]. The experimental values for tetrahedral nitrides are as follows: $E_{g}^{exp}\left(w-BN\right)=6.8\;eV$ [40], $E_{g}^{exp}\left(AlN\right)=6.09\;eV$ [41], $E_{g}^{exp}\left(GaN\right)=3.47\;eV$ [42,43], $E_{g}^{exp}\left(InN\right)=0.70\;eV$ [39,44,45,46]. The theoretical values are $E_{g}^{th}\left(BN\right)=6.77\;eV$, $E_{g}^{th}\left(AlN\right)=6.19\;eV$, $E_{g}^{th}\left(GaN\right)=3.41\;eV$, and $E_{g}^{th}\left(InN\right)=0.65\;eV$ [47]. In the case of hexagonal boron nitride (h-BN), the experimental bandgap is still being discussed, as it is unclear whether it is direct or indirect. Thus, this value should be treated with caution: $E_{g}^{th}\left(h-BN\right)=6.42\;eV$ [48]. Graphene has no bandgap in standard AB stacking. In general, the models used in theory recover experimental data reasonably well.

## 3. Results

### 3.1. Tetrahedral Nitrides

The wurtzite nitrides belong to two groups: BN and AlN, which have no *d* orbitals, whereas GaN and InN have ten electrons in the *d* shell. Therefore, the electronic properties of AlN will be considered first, as this compound is widely used in optoelectronics where the electronic properties are of primary importance [1]. The basic properties include the magnitude and type of the bandgap, as well as the effective mass and degeneracy of both the valence band (VB) and conduction band (CB). These properties follow from the band diagram. Figure 1 shows the AlN band diagram obtained from ab initio calculations, as well as the properties relevant to bonding.

Based on this data, the nitrogen N2s states are separated from the other valence band (VB) states by approximately 7 eV, forming a separate lower subband. This is in agreement with the experimental data of Magnuson et al. [23]. The PDOS for aluminum at the same energy is an artifact of the projection. In fact, the calculation of the band diagram does not create a quantum state associated with aluminum. Thus, this is a typical artifact of the projections. All Al2s and Al2p states are hybridized, forming bonding states with the neighboring atom N2p states. Although some contribution of antibonding nature is present, the dominant overlap is bonding. Naturally, the bonding of the conduction band is much stronger than antibonding. These states are empty and, therefore, do not contribute to the energy of the system. The obtained bandgap is direct.

The important issue is the type of bonding. In fact four Al-N bonds are actually formed from four hybridized Al-2s-2p states and three N2p states. This is not standard, as these states have the same energy; therefore, they are not separated by the overlap. In fact, this is a purely quantum effect of the creation of resonant states, i.e., states of the same energy. This formulation originated in wave mechanics and was first identified by Kekule in the model of carbon bonding in benzene [51,52]. According to the model, carbon–carbon bonds in the ring are formed by six electrons. Due to the Hund rule, the electrons form spin pairs, resulting in three bonds. Due to the sixfold symmetry of the ring, the pair of electrons must be allocated to both bonds simultaneously. This is really the case, where the electron occupies both bonding states with equal probability, i.e., P=1/2. In terms of wave mechanics, the electron switches between resonant states. In quantum language, this corresponds to the above probability. Note that the result is independent of the temperature. This is the result of the existence of the coherent state in benzene [52]. The concept of resonant bonding was subsequently developed and applied to the description of other molecules. An important case is related to sulfur and oxygen bonds, i.e., ${SO}_{2}$, ${SO}_{2}^{-}$, $O_{3}$ [21,53,54,55,56,57]. It is worth stressing that the theoretical findings were confirmed by the experimental results [21,56].

Significant extension of this concept to a solid state was formulated by P. W. Anderson in a description of a superconductivity in copper oxide structures [58]. Subsequently, the concept was applied to other systems [54,59]. These states were most recently identified in the configuration of a N adatom at a Ga covered GaN(0001) surface [60]. It was shown that the N adatom, similarly to N atoms in the GaN lattice, creates separated states. Thus, the energy difference is too high to be incorporated into the same bonding states, and they remain separated. It was concluded that the states are occupied with the probability P=3/4. This enables tetrahedral bonding in AlN to be fulfilled. It is consistent with the electron counting rule (ECR) [61,62]. The number of the electron is eight. These electrons are distributed as follows (spin factor is used): two in 2Ns states and 4$\times \left(3/4\right)\times 2=6$. Therefore, all states in the VB are occupied, and the AlN is a wide bandgap semiconductor.

This interpretation of the bonding symmetry is confirmed by the plots of the density of states for the Al and N atoms in the AlN lattice, shown in Figure 2. The N2s orbital, separate in the lower VB subband, is close to spherical, indicating a virtual absence of the overlap with the other orbitals. In the case of the upper VB subband, the shape of the *N2pz* orbitals is asymmetric with respect to the *N* atom (with respect to reflection along the 0z axis), indicating the overlap with the Al atom. Similarly, the asymmetric shape of both N2px and N2py orbitals in the 0xy plane indicates overlap with the Al atoms in this plane. The orbitals are directed towards the Al atoms, which confirms the resonating character of the bonding. The Bader charges are relatively high: $q_{B}\left(Al,N\right)=\mp 2.89,$ which confirms a relatively large electron shift to the N orbitals.

This observation can be verified through a direct assessment of the bonds between neighboring N and Al atoms. Such calculations could be performed using the generalized valence bond (GVB) theory [63], or use of the COHP formulation directly [49,50]. To calculate the COHP integrals, all the atoms must be incorporated into a single cell. Therefore, the size of the simulation system must be increased. Figure 3 shows the simulated system and COHP diagrams for a N atom and its four nearest neighboring (nn) Al atoms.

**Figure 3 materials-18-02875-f003:**
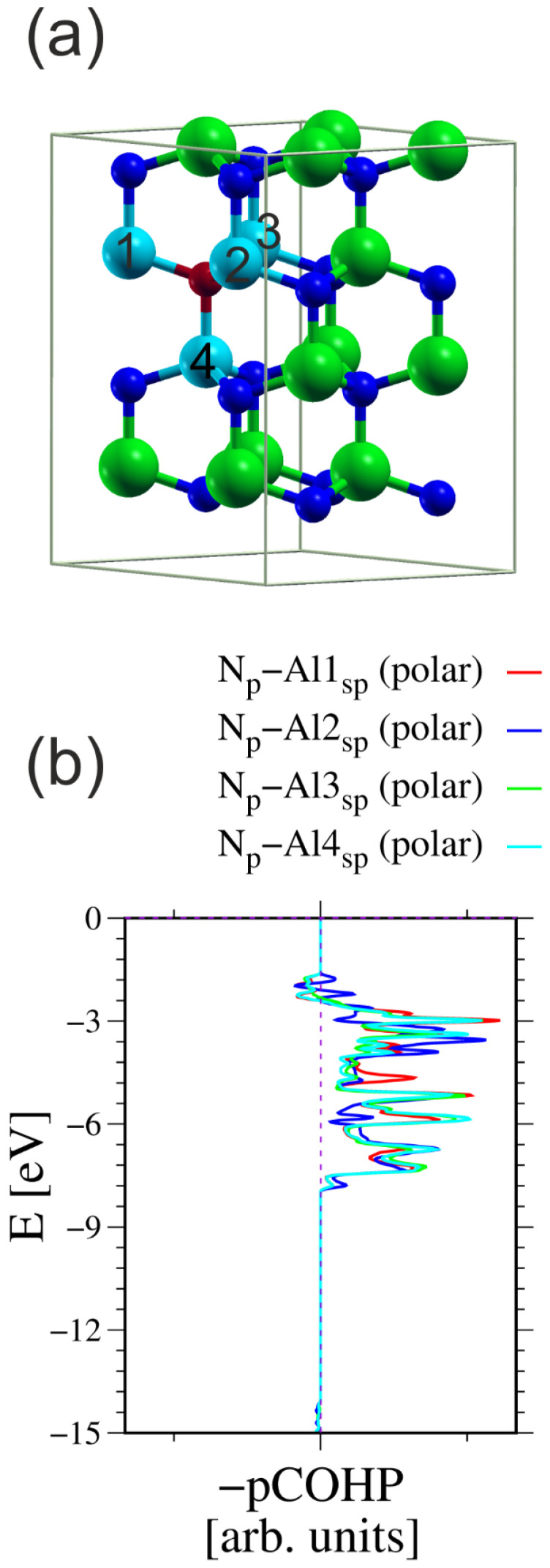
The AlN cell used for calculation of the COHP integrals between the N atom and its four nearest neighboring (nn) Al atoms: (**a**) simulated 2 × 2 cell having four Al-N double layers, in total 32 atoms; (**b**) COHP integrals for ensemble of three N2p orbitals and all sp^3^-hybridized Al orbitals [49,50]. The N atom used in the calculations is represented by a small red ball; the other N atoms are represented by small blue balls. The Al atoms used in the calculations are represented by large cyan balls; the other Al atoms are represented by large green balls. The numbers on Al atoms indicate those used in the calculation of the COHP integrals. The integration results are listed in Table 1.

From these values, it can be concluded that the bonding between all four atoms is identical within the precision of the calculations. Therefore, the system creates four bonds. Note that these bonds are not related by lattice symmetry. Additionally, the antibonding component of the charge is small, on the order of 2–3%. These values could be used for proper normalization to calculate the bond order of these bonds. In fact, the N atom contributes three electrons to the 2p orbitals, and the Ga atom also contributes also three electrons to the sp^3^-hybridized orbitals. Thus, the total number of electrons is six, which must be divided into four N-Al bonds. Using this normalization for an absolute value of the COHP integral, the bond order of these four bonds was calculated. The results are listed in Table 2.

These values confirm that the bonding in the AlN wurtzite semiconductor occurs via the formation of four resonant bonds. The antibonding fraction is most likely responsible for selecting the orbitals in the Kohn–Sham functional basis set and the projection of the COHP onto a grid.

A similar observation was made in the case of wurtzite boron nitride. As shown in Figure 4, the electronic properties of the system are, to some extent, close to the previous one. Again the *N2s* state is split, creating the lower VB subband. As before, the ghost state is observed in the boron PDOS. This state is not present in the band diagram, which confirms its ghost status. A significant difference is that the bandgap is indirect, with the conduction band minimum (CBM) at the K point.

The band width is higher than in the case of AlN. Similarly, the bandgap is wider. This is to be expected, given that the lattice constants of BN are 20% shorter. This is not much in terms of the distance, but according to a tight bonding approximation, it corresponds to a 50% increase in quantum overlap. This corresponds to the observed difference in VB widths of AlN and BN, respectively. Again, the bonding occurs between the ${sp}^{3}$-hybridized state of boron and N2p states of nitrogen. The bonding of the N2pxy states is mixed, whereas the bonding of the N2pz state is positive. The boron B2s state is relatively weakly bonded. The Bader charges are also high, at $q_{B}\left(B,N\right)=\mp 1.92,$ which confirms a relatively large electron shift from the B to the N orbitals.

A quite different picture emerges for GaN and InN. These metals have ten *d* electrons that contribute to the bonding. Figure 5 shows the electronic properties of GaN. As can be seen, the direct bandgap is wide, as expected.

As can be seen, the two VB subbands are formed: the lower one due to N2s and Ga3d orbitals, and the upper one related to ${sp}^{3}$-hybridized Ga and N2p orbitals. The subbands are separated by a 4 eV gap, which is comparable to the GaN bandgap. These results are fully compatible with the experimental data obtained by Magnuson for GaN [24]. The overlap is predominantly positive, with a small contribution from the N2s and Ga3d overlaps. In the upper VB subband, the higher part is dominated by Ga4pxy orbitals, as known from other ${sp}^{3}$-hybridized semiconductors. This is confirmed by the plots of the real part of the wavefunction in the GaN lattice at Γ point, shown in Figure 6. As shown in Figure 6a,b, the wavefunctions of the N2s state are distorted, showing considerable overlap with the Ga3d states. Again, the states N2p are asymmetric and directed towards the Ga atoms, which confirms the overlap with the neighboring Ga orbitals in the upper VB subband. Thus, the resonant bonding in the upper VB subband is confirmed. The Bader charges for the metal–nitrogen charge transfer are reduced, as compared to the previous cases: $q_{B}\left(Ga,N\right)=\mp 1.62,$ which confirms smaller electron shift to N orbitals.

Finally, indium nitride is also analyzed. In some respects, the compound is similar to the previous one, i.e., GaN. The difference is that it has a relatively narrow bandgap, which is typical of other III–V semiconductors but not of the tetrahedral nitrides. This low bandgap is related to the overlap of indium *In5s* orbitals and nitrogen N2s orbitals. Consequently, indium nitride exhibits the typical behavior of standard, fully ${sp}^{3}$ bonded III–V semiconductors. The extension of this state is responsible for a reduced InN bandgap. Accordingly, Kane theory predicts a small effective mass in the conduction band [64].

Nevertheless, these properties demonstrate striking similarities. Once again, the two VB subbands are formed: the lower one due to N2s and In4d orbitals, and the upper one related to ${sp}^{3}$-hybridized In and N2p orbitals. The subbands are separated by a 4 eV gap, which is comparable to the GaN gap. The overlap is predominantly bonding (positive), with a small contribution from the N2s and In4d overlaps. As is the case with other sp^3^-hybridized semiconductors, the higher part of the upper VB subband is dominated by In5pxy orbitals. This is confirmed by the plots of the states of the In and N atoms in an InN lattice, as shown in Figure 7. Bader charges are even smaller: $q_{B}\left(In,N\right)=\mp 1.40,$ which confirms a reduced electron shift to N orbitals, i.e., lower ionicity.

In summary, it is important to note the remarkable similarities between all wurtzite nitride semiconductor systems, regardless of their atomic states. The key point is the separation of the N2s states, which contribute to the lower VB subband. The crystal coherence is related to the upper subband, which enforces tetrahedral symmetry.

### 3.2. The Hexagonal Boron Nitride

In addition to the tetrahedral wurtzite nitride, boron nitride has an additional, more stable lattice, which is a layered 2D structure. Figure 8 shows the electronic properties of this system. As can be seen, the band structure is different. The bandgap is wide and indirect. The CBM and VBM are in the K point in the momentum space. Both bands are flat close to the extrema; thus, the effective masses of the electrons and holes are large. The bandgap is limited by the states created by overlap of the $N{2p}_{z}$ and $B{2p}_{z}$ orbitals, i.e., by the π states. The overlap between these states is large, indicating a substantial contribution from the π states to the bonding. In fact, π states are a major factor contributing to the stability of hexagonal phase. At the Γ point, the dispersions are opposite for π and σ states, which correspond to the energy minimum and maximum, respectively. The latter is typical of a tetrahedral valence band. These quantum states originate from the overlap of the in-plane $N{2p}_{xy}$ and $B{2p}_{xy}$. In general, the overlap of both types of bonding is indicated by the bandwidth and the COHP value. The magnitude is of the same order, suggesting that these two bands have a similar bonding strength. The theory of bonding originally proposed by Kane [64] and later applied to the general treatment of ${sp}^{3}$ bonded semiconductors by Harrison [21], can be understood in terms of energies of the bonding and antibonding states:(1)$$E_{b,a}=\left(E_{m}+E_{n}\right)/2\mp \;\sqrt{V_{2}^{2}+V_{3}^{2}}$$
where (*b*,*a*) indices denote bonding and antibonding states, respectively. The $V_{2}$ and $V_{3}$ are covalent and ionic strength, respectively. In the Harrison formulation, the ionic strength is defined based on the Hartree–Fock values calculated by Mann [57] as $V_{3}\;\equiv \;\left(E_{m}-E_{n}\right)/2$, using the metal (*m*)–nonmetal (*n*) energy difference. The covalent bonding is defined using effective Harrison parameters $\eta_{ijk}$, as $V_{2}\;\equiv \;\eta_{ijk}\hbar^{2}/md^{2}$. In the case of h-BN, both contributions are significant: ionic bonding is due to large metal–nitrogen energy difference, and covalent bonding is due to the small bond length.

Figure 9 provides further confirmation of the type of the bonding, showing plots of the wavefunctions for the upper and lower VB subbands. The data presented shows that the lower VB subband N2s orbitals create bonding. Similarly, the upper 2Npxy orbitals create bonding, which is visible from the B-N asymmetry. The N2pz orbitals form a π-type bond with high overlap that is parallel to the plane, and the wavefunction changes sign below and above the BN plane.

These data show that the separation of nitrogen N2s orbitals is preserved with the change of the lattice symmetry. In the case of hexagonal BN, the N2p orbital bonding is shifted to π-type bonding. The Bader charges for the B-N are relatively high: $q_{B}\left(B,N\right)=\mp 1.98,$ which confirms relatively large electron shift to the N orbitals.

These differences can be analyzed using COHP procedures in analogy to the above-considered wurtzite AlN [49,50]. Figure 10 shows the simulated system and COHP diagrams for a N atom and its three nearest neighboring (nn) B atoms.

**Figure 10 materials-18-02875-f010:**
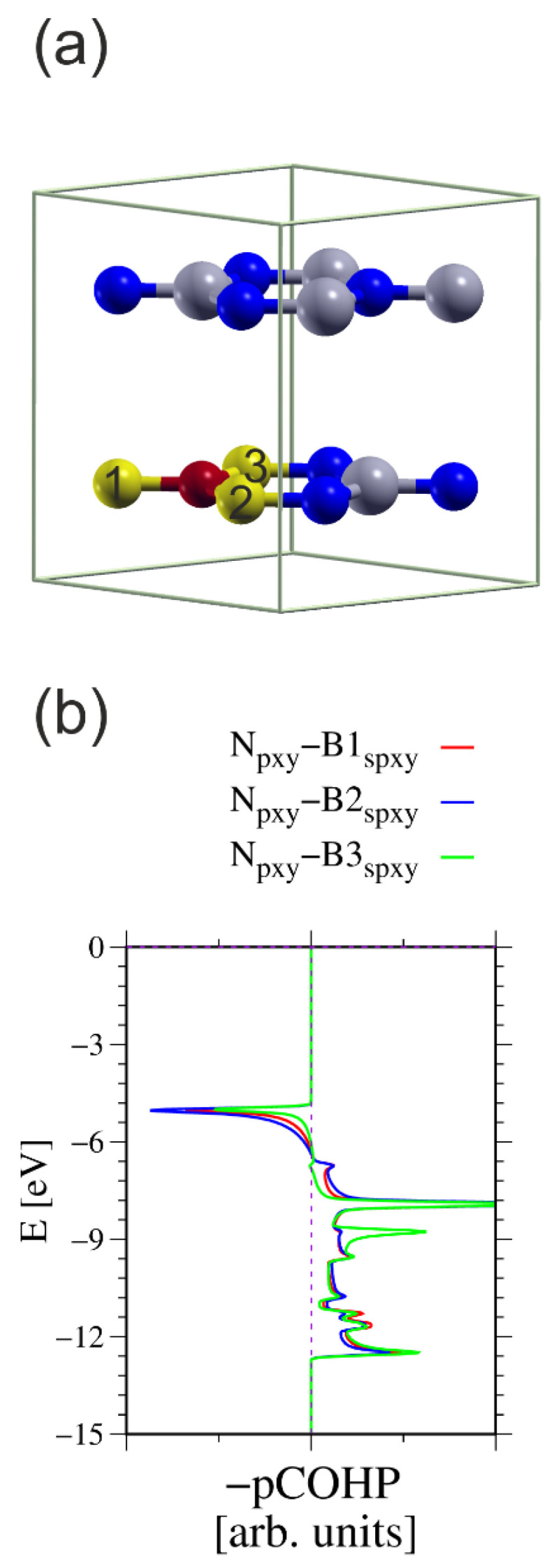
The h-BN cell used for calculation of the COHP integrals between the N atom and three nearest neighboring (nn) B atoms: (**a**) simulated 2 × 2 × 1 cell having two B-N layers, in total 16 atoms; (**b**) COHP integrals for ensemble of three N2p orbitals and all sp^2^-hybridized B orbitals [49,50]. The N atom used in the calculations is represented by a small red ball; the other N atoms are represented by small blue balls. The B atoms used in the calculations are represented by large yellow balls; the other B atoms are represented by large green balls. The numbers on the B atoms indicate which atoms were used to calculate the COHP integrals. The energy overlaps are listed in Table 3.

These data suggest that the COHP procedure results in less-reliable bond projections. The antibonding corresponds to a significant proportion of the electron density. This may be attributed to the basis, without a definitive answer to that aspect. This may be partially attributed to a different bonding type, both σ and π. This creates a difficulty in the proper projection of the molecular orbitals basis set. The above values could be used for proper normalization to calculate the bond order of these bonds. In fact, the N atom contributes three electrons to the 2p orbitals, and the B atom also contributes three electrons to the sp^3^-hybridized orbitals. From this, the two electrons should be subtracted for π bonds. Thus, the total number of electrons is four, which must be divided into three N-B bonds. The bond order of these three bonds was calculated using this normalization for the absolute value of the COHP integral. The results are listed in Table 4.

These values confirm that the bonding in the BN wurtzite semiconductor possibly occurs via the formation of three resonant bonds. The antibonding fraction most likely results from the selection of the orbitals in the Kohn–Sham functional basis set, connected with the different bonding type and the projection errors.

## 4. The Graphene

In the light of the dominant role of the large ionicity of the group III nitrides measured using the Harrison scale based on the Hartree–Fock values, which are much higher than those of other compounds [21,65,66,67] such as the arsenides, phosphides, and antimonides, an additional aspect of the separation of the nitrogen N2s orbitals is essential for the transition from the cubic zinc blende structure to the hexagonal wurtzite structure [68]. This is correlated with the emergence of spontaneous polarization in the wurtzite structures, which occurs when electrons shift toward nitrogen atoms [69]. An opposite effect has also an interesting aspect, i.e., the removal of ionicity in carbon compounds. This is of particular interest because the ionic contribution to the polarization is totally eliminated by virtue of the lattice symmetry. Therefore, this move leads to complete removal of the ionic tendency. This is confirmed by the Bader charges that are close to zero: $q_{B}\left(C,C\right)=\mp 0.09$. This small value is essentially zero, as this nonzero value stems from the selection of the wavefunction basis set. The effect is analogous to the zero polarization structure employed by Dreyer et al. [70]. Additionally, the absence of this difference leads to the closure of the gap, as already was identified in graphene because the bond energy is $E_{b,a}=E_{c}\mp \;V_{2}\;$. This is the standard solution used in the tight binding theory of the Dirac states in graphene [71]. Consequently, there is no bandgap and the dispersion relation is linear in the K point. This branch is clearly visible in Figure 11, where the graphene band structure is plotted. In fact, depending on the stacking, the two carbon atoms are inequivalent, as there is a small interaction between the carbon layers. In the case of AA stacking, the dispersion relation is strictly linear, whereas in the other AB and ABC stackings, a small gap is opened [72]. Nevertheless, due to large distance between graphene planes, the interaction is small, which could open the gap by a few meV. The Dirac states originate from the overlap of neighboring ${C2p}_{z}$ orbitals, which gives rise to π states.

In addition, the other bands are shown. The band structure is similar to that of hexagonal BN, except that the gap closes at the K point. Consequently, the ${sp}^{2}$ bonding states in VB are shown without any gap between the C2s and C2p states. Therefore, these states are hybridized. This is confirmed by the linear dependence observed in the COHP analysis of these states.

This type of the bonding is confirmed by the wavefunctions plots of the VB subbands, shown in Figure 12. The C2s orbital data shows significant overlap, indicating a substantial contribution to bonding overlap in accordance with the considerable dispersion of these states. In parallel, the 2Cpx and 2Cpy orbitals exhibit significant asymmetry, which is associated with bonding to neighboring C atoms. Additionally, π orbitals exhibit significant overlap in a plane parallel to the C plane. The wavefunction changes the sign across the C plane in according to the type of the bonding.

In summary, the bonding in graphene differs due to the absence of the ionic contribution that creates a Dirac-type band in the bandgap that is linear in the vicinity of the K point and is paramount in the graphene literature [71,72]. The bonding in the C plane is ${sp}^{2}$-hybridized.

## 5. Summary

A comprehensive ab initio study of the bonding of the nitride and graphene families was reported. This has altered the overall picture of the bonding in these structures. The importance of these findings can be assessed by direct comparison of the state of the art before and after the publication of this paper.

The bonding of a tetrahedral family was considered similar to that of other tetrahedral semiconductors, i.e., nitrides should be ${sp}^{3}$-hybridized orbitals bonded, like cubic zinc blende phosphides, arsenides, and antimonides. The results of Magnuson et al. were essentially ignored [23,24]. The bonding of hexagonal BN was considered to be partially by σ—${sp}^{2}$-hybridized orbitals and partially by $\pi \;p_{z}$ orbitals. The bandgap was opened due to the difference between the B and N atoms.

The results obtained in this paper indicate that the dominant feature of a bonding in nitrides is the separation of the N2s orbitals, which gives rise to two separate subbands: a lower, created by N2s orbitals with the overlap with Ga3d or In4d; and a higher one, created by the overlap of hybridized *sp* metal and N2p orbitals. This is in accordance with the measurements of Magnuson et al. [23,24]. The bonding of nitrides by a metal atom is tetrahedral ${sp}^{3}$, and hexagonal BN by ${sp}^{2}$-hybridized orbitals. The nitrogen bonds via resonant *N2p* states form four bonds in the tetrahedral and three bonds in the hexagonal BN. This is confirmed by the calculation of the bond orders using the COHP formalism, which yields noninteger values. The π-type bonding of nitrogen and boron 2pz orbitals creates additional bonding, which separates into a bonding and an antibonding state, opening the bandgap of a hexagonal boron nitride. The bonding in a graphene is different: it includes ${sp}^{2}$-hybridized orbitals that create bonds in the C plane. An additional π-type bond is formed by the C2pz orbitals out of the C plane. All these orbitals do not have the ionic contribution; therefore, the gap is closed by the Dirac-type states, having linear dispersion close to the *K* point.

Following publication, the state of the art changed due to the determination of the bonding of the nitrides that have complex character: the separation of two subbands and the resonant bonding by nitrogen states. The hexagonal structures (BN and graphene) have an additional bonding due to the π-type bonds. The bonding in a graphene is different; it is symmetric with no ionic contribution. Therefore, the covalent bonding model applies to graphene.

## Figures and Tables

**Figure 1 materials-18-02875-f001:**
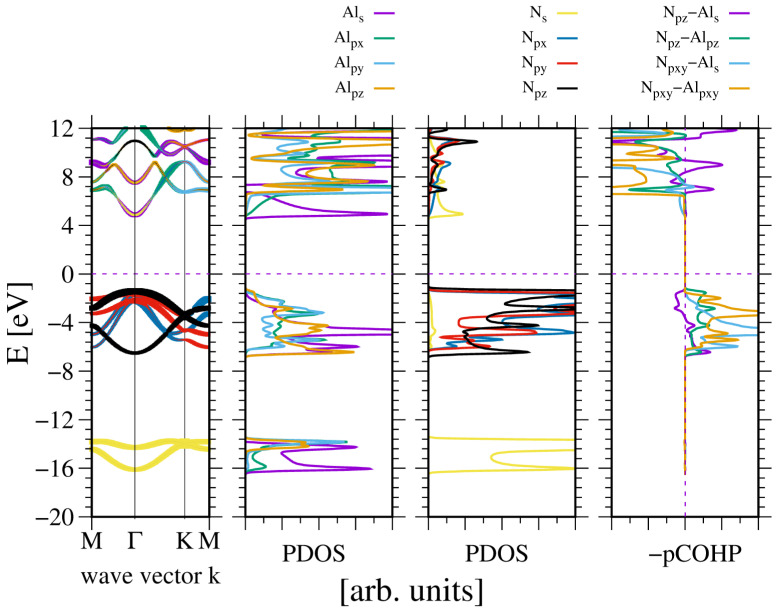
The electronic properties of bulk aluminum nitride (AlN). The panels represent, from the leftmost side: band diagram, projected density of states (PDOS) of the Al (**left**) and N (**right**) atoms; and the rightmost panel—projected Crystal Orbital Hamilton Population (COHP) [49,50]. The COHP data correspond to the pair of nearest neighboring (nn) N and Al atoms. The color codes are the same for the PDOS and the band diagram. The Fermi energy is set to zero. Positive COHP values correspond to the bonding overlap.

**Figure 2 materials-18-02875-f002:**
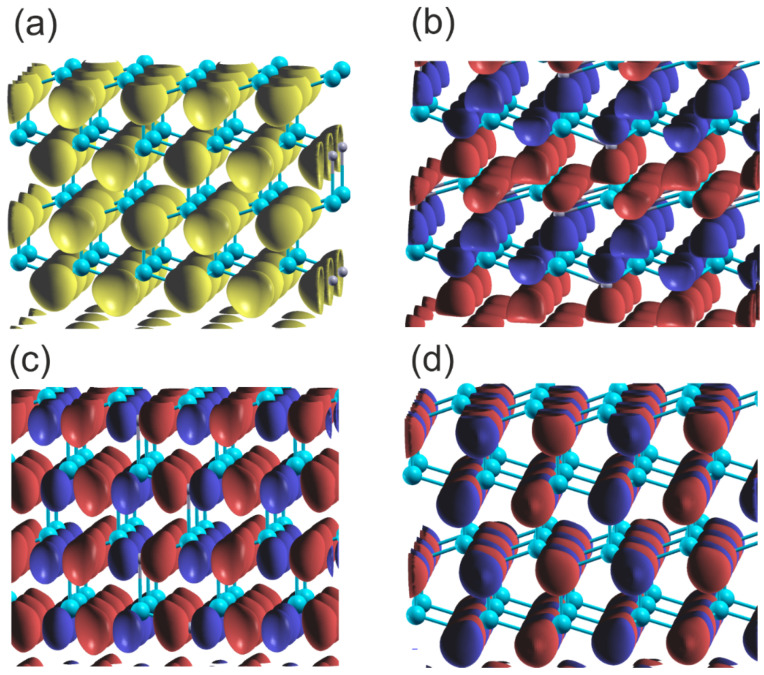
Plot of the real part of the wavefunctions (Re$\left(\Psi \right)$) of the orbitals of the upper and lower VB of wurtzite AlN at Γ point: (**a**) orbital N2s $\left(Max=0.16,\;Min=0.00,\;Isosurf=0.06\right)$, (**b**) orbital N2pz $\left(Max,Min=\pm 0.24,\;Isosurf=\pm 0.08\right)$, (**c**) orbital N2py $\left(Max,Min=\pm 0.37,\;Isosurf=\pm 0.08\right)$, (**d**) orbital N2px $\left(Max,\;Min=\pm 0.37,\;Isosurf=\pm 0.08\right)$. The blue and red colors represent the positive and negative parts of the function of N2p orbitals; in the case of 2Ns orbitals the only single part is present, which is marked in yellow. The cyan and gray balls represent Al and N atoms, respectively.

**Figure 4 materials-18-02875-f004:**
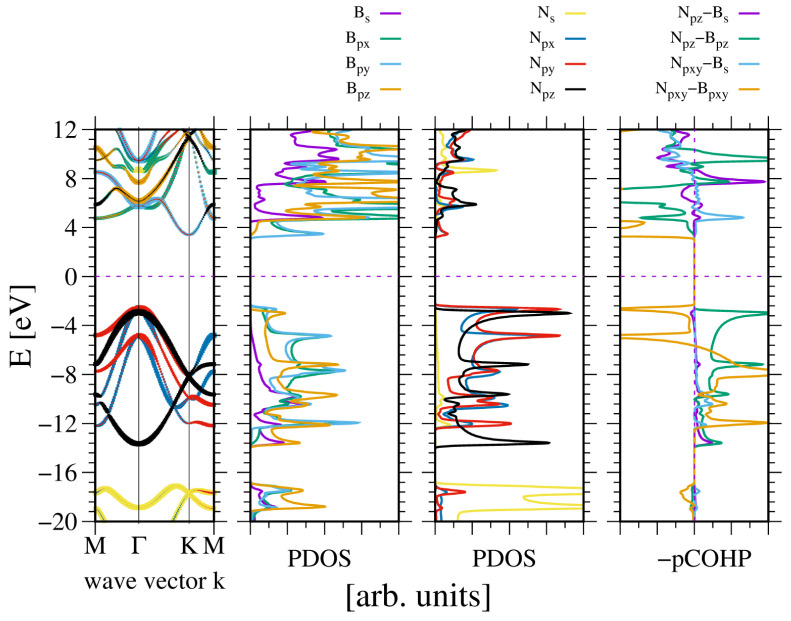
The electronic properties of bulk wurtzite boron nitride (BN). The panels represent, from the leftmost: band diagram, projected density of states (PDOS) of the B (**left**) and N (**right**) atoms; and the rightmost panel—Crystal Orbital Hamilton Population (COHP) [49,50]. The COHP data correspond to the pair of nearest neighboring N and B atoms. The color codes are the same for the PDOS and the band diagram. The Fermi energy is set to zero. Positive COHP values correspond to the bonding overlap.

**Figure 5 materials-18-02875-f005:**
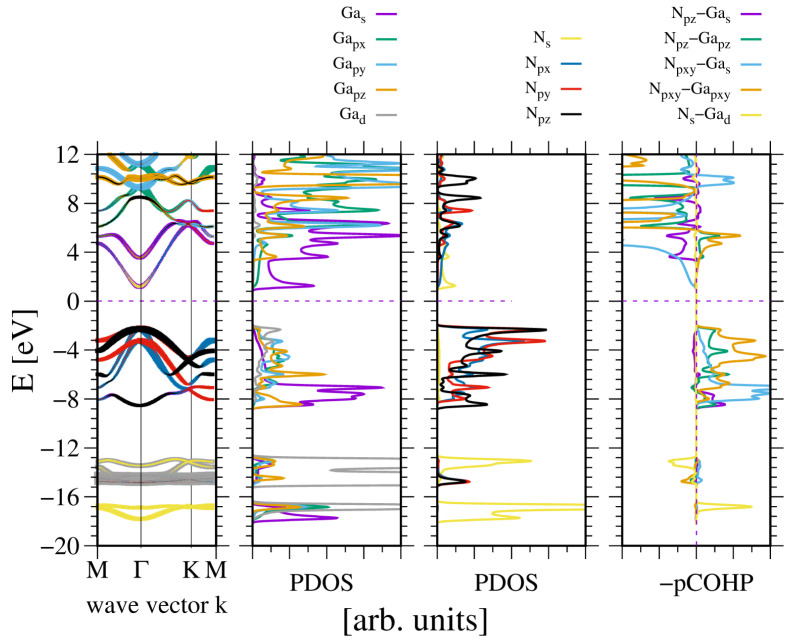
The electronic properties of bulk wurtzite gallium nitride (GaN). The panels represent, from the leftmost: band diagram, projected density of states (PDOS) of the Ga (**left**) and N (**right**) atoms; and the rightmost panel—Crystal Orbital Hamilton Population (COHP) [49,50]. The COHP data correspond to the pair of nearest neighboring N and Ga atoms. The color codes are the same for the PDOS and the band diagram. The Fermi energy is set to zero. Positive COHP values correspond to the bonding overlap.

**Figure 6 materials-18-02875-f006:**
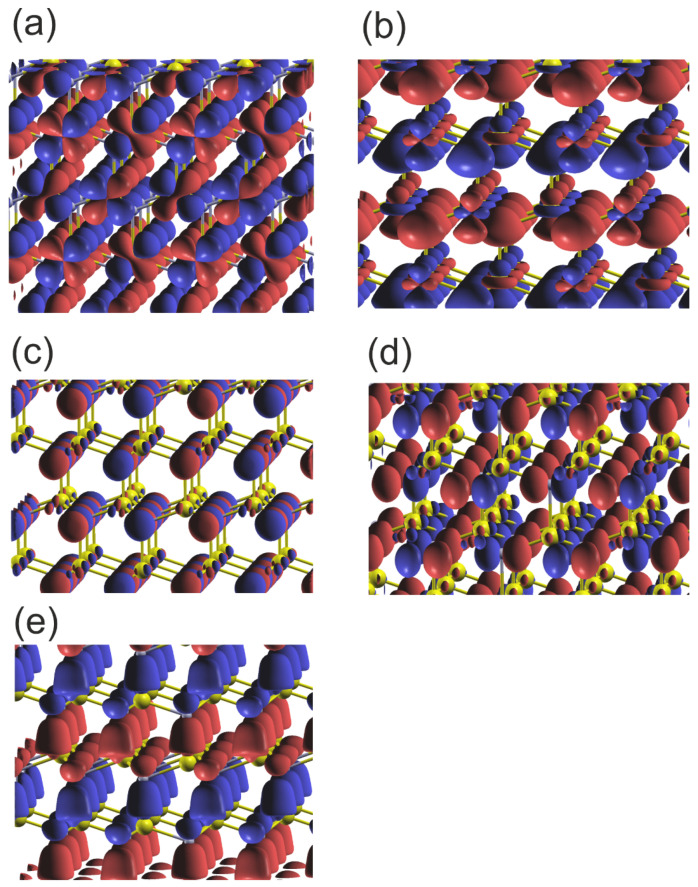
Plot of the real part of the wavefunctions (Re$\left(\Psi \right)$) of the orbitals of the upper and lower VB of wurtzite GaN at Γ point: (**a**) orbital Ga3d_yz $\left(Max=0.73,\;Min=-0.87,\;Isosurf=\pm 0.015\right)$; (**b**) orbitals N2s-Ga3d_z2 $\left(Max=0.73,\;Min=-0.73,\;Isosurf=\pm 0.06\right)$; (**c**) orbital N2px $\left(Max,\;Min=\pm 0.24,\;Isosurf=\pm 0.08\right)$; (**d**) orbital N2py $\left(Max,\;Min=\pm 0.37,\;Isosurf=\pm 0.08\right)$; (**e**) orbitals N2pz $\left(Max,\;Min=\pm 0.37,\;Isosurf=\pm 0.06\right)$. The colors represent the positive and negative components of the function of the N2p orbitals; in the case of the 2Ns orbitals the only single part is present. The yellow and gray balls represent Ga and N atoms, respectively.

**Figure 7 materials-18-02875-f007:**
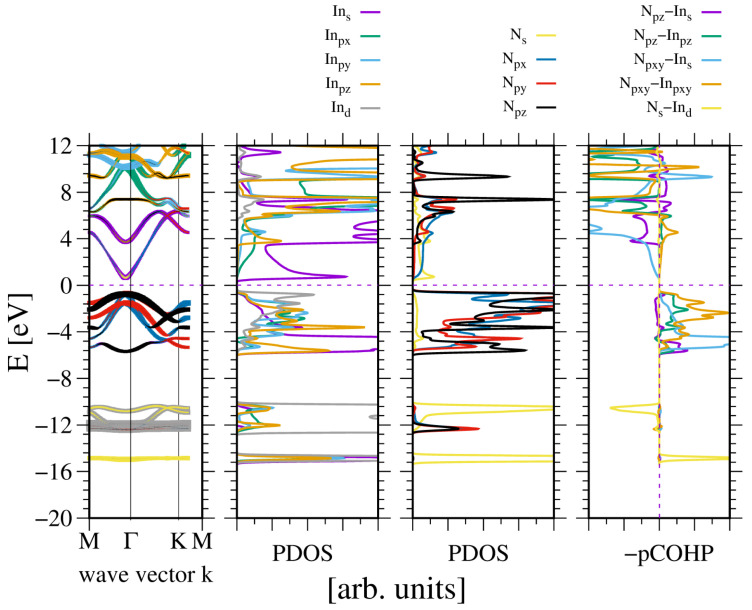
Electronic properties of the bulk wurtzite indium nitride (InN). The panels represent, from the leftmost: band diagram, projected density of states (PDOS) of the In (**left**) and N (**right**) atoms; and the rightmost panel—Crystal Orbital Hamilton Population (COHP) [49,50]. The COHP data correspond to the pair of nearest neighboring N and In atoms. The color codes are the same for the PDOS and the band diagram. The Fermi energy is set to zero. Positive COHP values correspond to the bonding overlap.

**Figure 8 materials-18-02875-f008:**
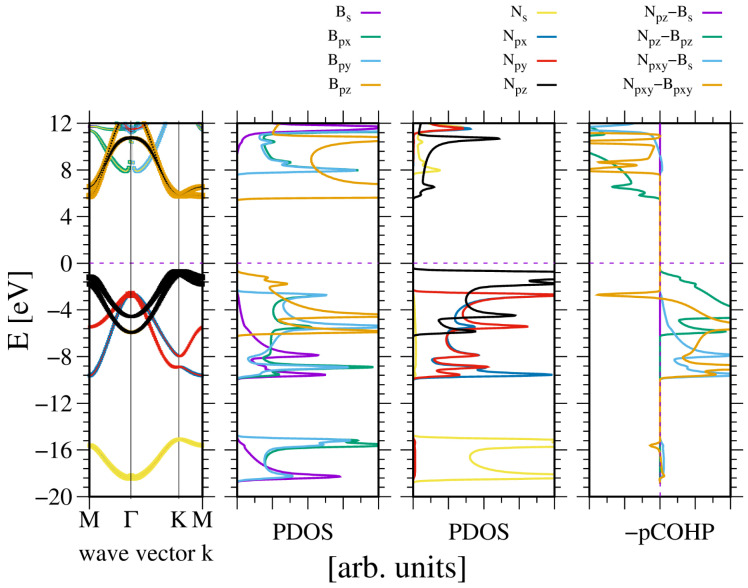
Electronic properties of periodic hexagonal boron nitride (h-BN). The panels represent, from the leftmost: band diagram, projected density of states (PDOS) of the B (**left**) and N (**right**) atoms; and the rightmost panel—Crystal Orbital Hamilton Population (COHP) [49,50]. The COHP data correspond to the pair of nearest neighboring N and B atoms. The color codes are the same for the PDOS and the band diagram. The Fermi energy is set to zero. Positive COHP values correspond to the bonding overlap.

**Figure 9 materials-18-02875-f009:**
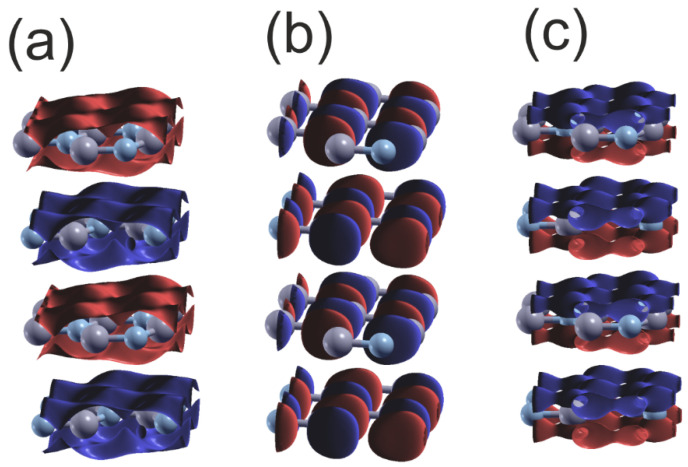
Plot of the real part of the wavefunctions (Re$\left(\Psi \right)$) of the orbitals of the upper and lower VB of hexagonal BN at Γ point: (**a**) orbital N2s $\left(Max,\;Min=\pm 0.17,\;Isosurf=\pm 0.05\right)$; (**b**) orbital N2py $\left(Max,Min=\pm 0.4,\;Isosurf=\pm 0.05\right)$; (**c**) orbital N2pz $\left(Max,Min=\pm 0.29,\;Isosurf=\pm 0.08\right)$. The colors represent the positive and negative parts of the function of N2p orbitals; in the case of 2Ns orbitals only the single part is present, and the sign changes for the different layers. The gray and bluish–gray balls represent B and N atoms, respectively.

**Figure 11 materials-18-02875-f011:**
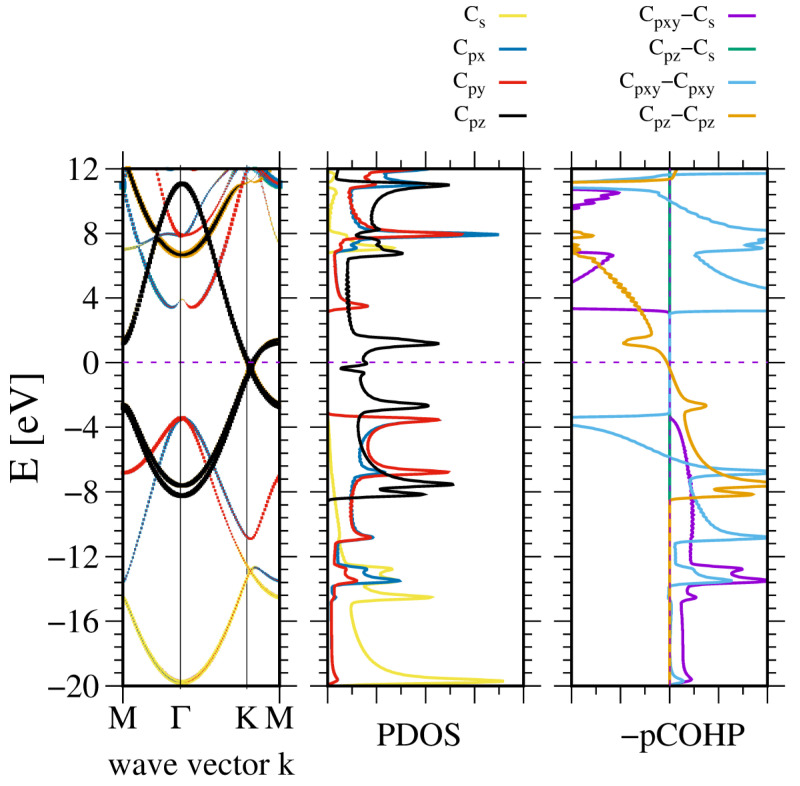
The electronic properties of AB stacked graphene. The panels represent, from the (**leftmost**): band diagram, projected density of states (PDOS) of the C atom (**middle**); and the (**rightmost**) panel—Crystal Orbital Hamilton Population (COHP) [49,50]. The COHP data correspond to the pair of nearest neighboring N and B atoms. The color codes are the same for the PDOS and the band diagram. The Fermi energy is set to zero. Positive COHP values correspond to the bonding overlap.

**Figure 12 materials-18-02875-f012:**
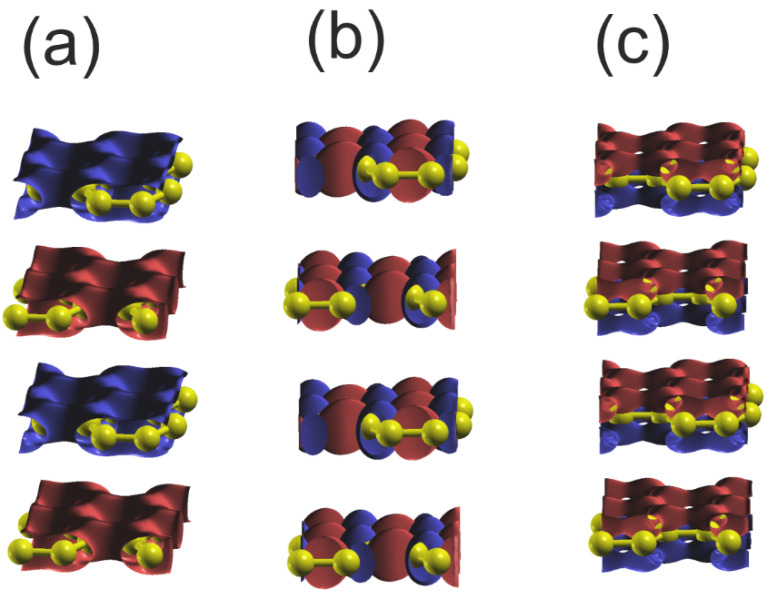
Plot of the real part of the wavefunctions (Re$\left(\Psi \right)$) of the orbitals of the upper and lower VB of graphene at Γ point: (**a**) orbital C2s $\left(Max,Min=\pm 0.13,\;Isosurf=\pm 0.08\right)$; (**b**) orbital C2py $\left(Max,Min=\pm 0.29,\;Isosurf=\pm 0.05\right)$; (**c**) orbital C2pz $\left(Max,Min=\pm 0.19,\;Isosurf=\pm 0.08\right)$. The colors represent the positive and negative parts of the C2p orbital function. In the case of the 2Cs orbital, only the single part is present, and the sign changes for the different layers. The yellow balls represent C atoms.

**Table 1 materials-18-02875-t001:** The calculated COHP bonding (positive) and antibonding (negative) of AlN (in arb units).

Atoms	$\langle H\left(N-Al\right)\rangle \;Bonding$	$\langle H\left(N-Al\right)\rangle \;Antibonding$	$\langle H\left(N-Al\right)\rangle \;Total$
N-Al1	1.753	−0.057	1.867
N-Al2	1.707	−0.018	1.743
N-Al3	1.685	−0.038	1.761
N-Al4	1.691	−0.053	1.796

**Table 2 materials-18-02875-t002:** Assessed and calculated bond order and bond lengths in AlN.

Atoms	Bond Order—Assessed	Bond Order—Calculated	$\mathrm{Bond}\;\mathrm{Length}\;\left(Å\right)$
N-Al1	0.75	0.704	1.896
N-Al2	0.75	0.735	1.896
N-Al3	0.75	0.718	1.896
N-Al4	0.75	0.706	1.910

**Table 3 materials-18-02875-t003:** The calculated in-plane COHP bonding (positive) and antibonding (negative) of h-BN (in arb. units).

Atoms	$\langle H\left(N-B\right)\rangle \;Bonding$	$\langle H\left(N-B\right)\rangle \;Antibonding$	$\langle H\left(N-B\right)\rangle \;Total$
N-B1	1.545	−0.321	2.188
N-B2	1.409	−0.468	2.346
N-B3	1.693	−0.189	2.072

**Table 4 materials-18-02875-t004:** Assessed and calculated bond order and bond lengths of in-plane h-BN.

Atoms	Bond Order—Assessed	Bond Order—Calculated	$\mathrm{Bond}\;\mathrm{Length}\;\left(Å\right)$
N-B1	0.66	0.530	1.451
N-B2	0.66	0.613	1.451
N-B3	0.66	0.450	1.451

## Data Availability

The raw data supporting the conclusions of this article will be made available by the authors on request.

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
