# Peer review of "Ab Initio Elucidation of the Nature of the Bonding of Tetrahedral Nitrides (BN, AlN, GaN, and InN), Hexagonal BN, and Graphene"

_materials, 2025, doi:10.3390/ma18122875_

Round 1
Reviewer 1 Report
Comments and Suggestions for Authors
The manuscript entitled “Ab initio elucidation of the nature of the bonding of tetrahedral nitrides (BN, AlN, GaN and InN), hexagonal BN and graphene”. The manuscript addresses a relevant topic: the nature of bonding in tetrahedral and hexagonal nitrides as well as in graphene, using DFT. The topic is timely, especially considering the growing importance of III-nitrides in optoelectronics and the continuing interest in 2D materials like h-BN and graphene. However, the manuscript presents some issues that must be addressed before it can be considered for publication. My primary concerns are as follows:
- The computational setup lacks adequate justification. In particular, the use of a 1×1×1 Monkhorst-Pack grid is extremely questionable for these materials and is likely insufficient to achieve convergence. The use of GGA-1/2 is justified but not clearly implemented. There is a lack of detailed convergence testing or comparison with more accurate methods (e.g., hybrid functionals or GW). The authors should include convergence tests for both issues to validate their computational choices.
- The study claims to provide a new picture of bonding via "resonant nitrogen states" in tetrahedral nitrides and a comparison to bonding in h-BN and graphene. This statement implies that the work has originality and scientific merit, but is not rigorous defined. The analogy with benzene and superconducting oxides is speculative and lacks solid justification in the context of III-nitrides. A quantitative analysis, like Bader charges, ELF, or bond orders, can help to support this statement.
- The use of COHP is welcome, but the analysis is superficial. COHP plots are mentioned but not deeply discussed or compared across materials. There is no energy-resolved quantification of bonding/antibonding contributions near the Fermi level.
- Wavefunction isosurfaces are shown but lack standardization and scale. Without proper normalization or comparison, their interpretation is subjective.
- Include comparative bonding analysis across systems (e.g., bond lengths, COHP integrals, Bader charges).
The manuscript contains numerous grammatical and syntactic errors, which hinder readability and scientific clarity. I strongly recommend a thorough revision of the text by a professional English-language editor. Improving the grammar and overall tone will greatly enhance the manuscript’s clarity and presentation.
Author Response
The reviewer comments are written in standard text, our response are in italics .
Reviewer #1
- The computational setup lacks adequate justification. In particular, the use of a 1×1×1 Monkhorst-Pack grid is extremely questionable for these materials and is likely insufficient to achieve convergence. The use of GGA-1/2 is justified but not clearly implemented. There is a lack of detailed convergence testing or comparison with more accurate methods (e.g., hybrid functionals or GW). The authors should include convergence tests for both issues to validate their computational choices.
This is an error in writing for which we apologize. The grid was . The error is corrected. We have also done convergence testing with respect of the number of k-points and this grid was enough for lattice constants and energy gap in GGA-1/2. There is no HSE approximation implemented in Siesta. Actually the predictions of GGA-1/2 are more precise for nitrides in terms of experimental values than the hybrid functional and the error is of the order of GW calculations.
- The study claims to provide a new picture of bonding via "resonant nitrogen states" in tetrahedral nitrides and a comparison to bonding in h-BN and graphene. This statement implies that the work has originality and scientific merit, but is not rigorous defined. The analogy with benzene and superconducting oxides is speculative and lacks solid justification in the context of III-nitrides. A quantitative analysis, like Bader charges, ELF, or bond orders, can help to support this statement.
The application of bond order is great idea and we are grateful to the Referee. Bond orders indicate on number of electrons in bonding and antibonding states. They are useful for indicating bonding in many molecules. We have found rich literature on this subject which is cited now. So these calculations are added in new version. And these numbers are noninteger for both tetrahedral and hexagonal nitrides. We have added also COHP diagrams that were used to assess the electrons in bonding and antibonding states.
In fact reference to Anderson work is important as this is the first application of the resonant bonding in solid state physics.
We thank the Referee for these suggestions most sincerely.
- The use of COHP is welcome, but the analysis is superficial. COHP plots are mentioned but not deeply discussed or compared across materials. There is no energy-resolved quantification of bonding/antibonding contributions near the Fermi level.
Actually GVB and COHP are equivalent, but different approaches, as these are valence band and molecular orbital approaches. We added derived numerical values of the COHP of these bonds in the analysis of the above point. They are added in this context.
- Wavefunction isosurfaces are shown but lack standardization and scale. Without proper normalization or comparison, their interpretation is subjective.
This is standard, e.g. in Ref. 45, which is authorative study these number are not listed.
- Include comparative bonding analysis across systems (e.g., bond lengths, COHP integrals, Bader charges).
The bond length are added, similarly the COHP integral values. .
The English was corrected.

Reviewer 2 Report
Comments and Suggestions for Authors
The following comments should be addressed:
- Use schematic diagrams to illustrate the unconventional bonding model (e.g., visual comparison of standard sp³ bonding vs resonant bonding proposed).
- Provide a stronger justification for the 1×1×1 Monkhorst-Pack grid—this is unusually coarse and may raise concerns about convergence.
- Clarify why the GGA-1/2 approach was preferred over hybrid functionals (e.g., HSE06) which are more commonly used for accurate bandgap predictions.
- Improve figure captions to include brief descriptions of what each panel shows and how it supports the bonding model.
- The authors should briefly touch upon how annealing affects Al-N bonding and cite corresponding literature (10.3390/coatings10100954, 10.1039/C5RA04728E)
- Add insets or overlays in the PDOS plots to highlight subband separation and key orbital contributions.
- Where “ghost states” are mentioned (e.g., for B PDOS), explain their origin and implications more clearly.
- Clarify the distinction between resonant bonding in molecular chemistry (e.g., benzene) and in the solid-state systems considered here.
Author Response
The reviewer comments are written in standard text, our response are in italics .
Reviewer #2
- Use schematic diagrams to illustrate the unconventional bonding model (e.g., visual comparison of standard sp³ bonding vs resonant bonding proposed).
We are not sure how these diagrams should look like. In the cited literature no such diagrams exist.
- Provide a stronger justification for the 1×1×1 Monkhorst-Pack grid—this is unusually coarse and may raise concerns about convergence.
This is an error in writing for which we apologize. The grid was . The error is corrected. We have also done convergence testing with respect the number of k-points and this grid was sufficiently dense for lattice constants and energy gap in GGA-1/2.
- Clarify why the GGA-1/2 approach was preferred over hybrid functionals (e.g., HSE06) which are more commonly used for accurate bandgap predictions.
There is no HSE approximation implemented in Siesta. Actually the predictions of GGA-1/2 for nitrides are even closer to the experimental values.
- Improve figure captions to include brief descriptions of what each panel shows and how it supports the bonding model.
The captions were corrected.
- The authors should briefly touch upon how annealing affects Al-N bonding and cite corresponding literature (10.3390/coatings10100954, 10.1039/C5RA04728E)
The annealing affects the nitrides in general as they consist of highly volatile nitrogen which is likely to escape. This is well known problem for all nitrides, which is important for AlN, GaN and InN. Some remedies were found but they require use of heigh pressures on nitrogen and limited pressure of the metal. Thus the process is difficult to control. Therefore all nitrides are affected which may be source of some additional errors, The references are included. We thank the Referee for point the problem to us.
- Add insets or overlays in the PDOS plots to highlight subband separation and key orbital contributions.
The insets cannot be inserted as most of the figures area are covered by the lines.
- Where “ghost states” are mentioned (e.g., for B PDOS), explain their origin and implications more clearly.
The ghost states originate from projection procedure only. This is due to the fact that projection uses the projection on the basis that is minimal and is not orthogonal. There is no such states in the basis.
- Clarify the distinction between resonant bonding in molecular chemistry (e.g., benzene) and in the solid-state systems considered here.
In our opinion no distinction exist. In fact both are pure quantum effects, the fractional occupation of the states that can be translated into the resonance between wavefunctions.

Round 2
Reviewer 1 Report
Comments and Suggestions for Authors
The authors have adequately addressed the questions raised in the first report. I can recommend the publication of the manuscript.
Reviewer 2 Report
Comments and Suggestions for Authors
The authors have improved their paper, publication is now justified.